# A Novel Cold-Adapted and Salt-Tolerant RNase R from Antarctic Sea-Ice Bacterium *Psychrobacter* sp. ANT206

**DOI:** 10.3390/molecules24122229

**Published:** 2019-06-14

**Authors:** Yatong Wang, Yanhua Hou, Ping Nie, Yifan Wang, Xiulian Ren, Qifeng Wei, Quanfu Wang

**Affiliations:** Harbin Institute of Technology, School of Marine Science and Technology, Weihai 264209, China; wangyatong199311@163.com (Y.W.); 15104589482@163.com (P.N.); daid0101@163.com (Y.W.); renxiulian1229@163.com (X.R.); weiqifeng163@163.com (Q.W.)

**Keywords:** RNase R, cold-adapted, antarctic bacterium, sea-ice, homology modeling

## Abstract

A novel RNase R, *psrnr*, was cloned from the Antarctic bacterium *Psychrobacter* sp. ANT206 and expressed in *Escherichia coli* (*E. coli*). A bioinformatics analysis of the *psrnr* gene revealed that it contained an open reading frame of 2313 bp and encoded a protein (PsRNR) of 770 amino acids. Homology modeling indicated that PsRNR had reduced hydrogen bonds and salt bridges, which might be the main reason for the catalytic efficiency at low temperatures. A site directed mutation exhibited that His 667 in the active site was absolutely crucial for the enzyme catalysis. The recombinant PsRNR (rPsRNR) showed maximum activity at 30 °C and had thermal instability, suggesting that rPsRNR was a cold-adapted enzyme. Interestingly, rPsRNR displayed remarkable salt tolerance, remaining stable at 0.5–3.0 M NaCl. Furthermore, rPsRNR had a higher *k*_cat_ value, contributing to its efficient catalytic activity at a low temperature. Overall, cold-adapted RNase R in this study was an excellent candidate for antimicrobial treatment.

## 1. Introduction

Ribonuclease (Ribonuclease, RNase), an important nucleic acid hydrolase in organism, can hydrolyze the phosphodiester bonds among the nucleic acid residues of RNA and play a significant role in different stages of the cell metabolism [1,2]. In *Escherichia coli* (*E. coli*), eight distinct RNases involved in different aspects of RNA metabolism were identified. RNase R, originally termed as *vacB* in *Shigella flexneri* [3], was found in *Aeromonas hydrophila* [4], *Legionella pneumophila* [5], and *Pneumoniae* [6]. RNase R participates in an essential cell function besides its role in virulence [3]. It has been reported that RNase R can not only participate in controlling the type and quantity distribution of intracellular RNA but degrade different types of RNA [7]. RNase R’s interaction with ribosomal proteins might be a result of this enzyme’s involvement in the ribosome quality control [8,9]. A growing number of evidence has indicated that RNase R is a cold shock protein (CspA), and its expression regulation is closely related to microbial adaptation [10,11]. It is well-known that the recruitment of RNase R to the degradosome complex is related to its need for low temperatures during growth [12]. It has been discovered that RNase R in *Pseudomonas syringae* plays an important role in growth at low temperature [13]. More importantly, RNase R is necessary for maintaining the precise amount of active ribosomal complexes required for proper mRNA translation, and, accordingly, RNase R might be a promising target for antimicrobial treatment [4]. Due to the function of RNase R in RNA metabolism and biological activities [3,12], it might play an important role in the understanding of virulence associated with enterobacteria and have great value in antimicrobial treatment. 

The unique geographical and climatic characteristics of Antarctic sea-ice have produced a wealth of extreme microbial resources. In the long-term evolution, microbes gave birth to very unique genetic resources, genetic backgrounds, and metabolic characteristics. Recently, it has been demonstrated that RNA transcription factors and protein synthetic factors play a vital part in Antarctic bacterium low temperature adaptation mechanisms [14]. Due to it being key enzyme in the regulation of RNA metabolism system, a study concerning the structural and biological characteristics of RNase R from Antarctic organisms is imminent. The present study was aimed at conducting molecular and enzymatic characterizations of the cold-adapted RNase R to have an insight into its structural and catalytic properties. 

## 2. Results and Discussion

### 2.1. Identification of psrnr Gene

The *psrnr* gene was seen to be 2313 bp in length (GenBank accession numbers MK624989) and to encode a protein with 770 amino acids, while the length of the *rnr* gene from *Pseudomonas syringae* Lz4W was seen to be 2658 bp, encoding 885 amino acids [12]. Based on sequence alignments with the related RNase Rs, the residues (Thr 646, Phe 654, Leu 665 and His 667) were identified as the catalytic sites in the sequence of PsRNR (Figure 1). However, RNase R from *E. coli* DEC6A was seen to have two catalytic sites (Asp 272 and Asp 281) [15]. Besides, it was reported that two catalytic sites from *E. coli* were identified in Tyr 324 and Asp 280 [16]. Another RNase R from *E. coli* contained three catalytic sites (Asp 272, Asp 278, and Asp 280) [17]. In addition, the sequence homology showed that PsRNR displayed the highest similarity (90.07%) to RNase R from *Psychrobacter cryohalolentis*, followed by the RNase R from *Psychrobacter arcticus* (88.98%). 

### 2.2. Homology 3D Modeling and Structure Analysis

For the validation of the structural model, 95.89% of the residues had an averaged 3D-1D score ≥ 0.2. The model was validated by a Ramachandran plot analysis with 90.8% residues in the most favored regions. These parameters indicated that the structural model of PsRNR was well qualified. As shown in Figure 2, the structural model of PsRNR superimposed well with the homologous RNase R (EcRNase R, PDB: 5XGU). Furthermore, in comparing the structural features between PsRNR and EcRNase R, it can be seen that the former showed several cold-adapted characteristics (Table 1). Firstly, in comparison with its mesophilic homologues, PsRNR showed less electrostatic interactions, particularly hydrogen bonds and salt bridges, resulting in a decrease in stability and thermostability [18,19]. A similar study on cold-adapted chitinase also revealed that the reduction of electrostatic interactions contributed to its thermal instability [20]. Secondly, less hydrophobic interactions might also be advantageous to the structural flexibility of cold-adapted enzymes [21], and PsRNR possessed reduced hydrophobic interactions (447) in comparison with EcRNase R (543). Furthermore, PsRNR showed a lower Arg/(Arg + Lys) ratio, which manifested as increased conformational flexibility and low temperature catalytic competence [22]. Similar structural features were also obtained in the study of the cold-adapted esterase [23].

### 2.3. Expression, Purification and Enzyme Assays

In comparing BL21/pET-28a(+) with isopropyl β-D-thiogalactoside (IPTG) induction (Figure 3, Lane 2), one can see that the crude extract from the recombinant bacteria displayed a different band with an approximately molecular mass of 91.4 kDa (Figure 3, Lane 3, red arrow), indicating that PsRNR was expressed in *E. coli* BL21. The purified rPsRNR displayed a major band on SDS-PAGE (Figure 3, Lanes 4–6). The rPsRNR was purified approximately 5.58-fold with a recovery yield of 48.85%, and its specific activity was 115.60 μmol/min/mg; that of RNase R from *E. coli* was 13-fold with a recovery yield of 40% [7]. 

### 2.4. Site-Directed Mutagenesis

The activity of the mutant rPsRNR generated by site-directed mutagenesis was investigated. As shown in Figure 4, the loss of the enzymatic activity of three mutants T646A, F654A, and L665A, was 12%, 55%, and 50%, respectively, confirming that these residues might participate in the process of RNase R catalysis [24]. Importantly, the H667A mutant protein completely lost the activity toward yeast RNA, indicating that His667 was necessary for catalysis. Nevertheless, residues of D272 and D280 in *E. coli* RNase R were crucial for its catalytic activity [17], and the residue of D280 in RNase R played a key role in its RNase activity [16]. 

### 2.5. Biochemical Characteristics of rPsRNR

The optimal temperature of rPsRNR was 30 °C (Figure 5a), while the RNase R from the Antarctic bacterium *Pseudomonas syringae* showed maximal activity at 25 °C [12]. Furthermore, the maximal activity of an extracellular RNase from *Bacillus cereus* was at 60 °C [25]. Importantly, rPsRNR, which had lower activity than psychrophilic RNase R, retained 22.0–38.0% of its initial activity even at 0–10 °C [12]. For the enzymatic thermostability, the half-life of the rPsRNR activity was approximately 22 min at 50 °C, but rPsRNR was completely inactivated for 70 min at 50 °C (Figure 5b). In contrast, RNase R from *E. coli* lost approximately 80% of its full activity when incubated 5 min at 50 °C [12]. RNase II, an exoribonuclease homologous to RNase R, was most active at 50 °C for 5 min [7]. The optimum pH for rPsRNR activity was 6.0 (Figure 5c), while the iso-Ribonucleases from thermophilic fungus showed full activity at pH 3.0 [26]. In addition, RNase II from *E. coli* displayed a broad optimal pH range between 7.5 and 9.5 [7], and the activity of RNase R from *Mycoplasma genitalium* was maximum at pH 8.5; the activity dropped by about 20% at pH 7.5 and 9.0 [27]. As for the pH stability, rPsRNR maintained about 80.0% of its maximal activity over the pH range of 5.5 to 6.5 at 30 °C. (Figure 5d), suggesting that rPsRNR exhibited good stability under the weak acid condition. On the contrary, RNase HII from *Aeropyrum pernix* exhibited maximal activity at an alkaline pH [28]. This range of pH dependence for activity and stability means that rPsRNR could potentially contribute to the medical industry. The activity of rPsRNR was investigated in the presence of various concentrations of NaCl (Figure 5e). rPsRNR kept its highest activity in the presence of 1.5 M NaCl and kept over 85.0% of its activity at 0.5–3.0 M NaCl; it lost approximately 36.9% of its activity at the concentration of 4.0 M. This data reflects that rPsRNR features salt tolerance characteristics, which may be related to the high salt concentration in Antarctic sea ice. Based on rPsRNR’s activity on the effects of various compounds (Table 2), rPsRNR was strongly inhibited by 1 mM Ba^2+^, and the rPsRNR activity was not detected after being incubated in 5 mM Ba^2+^ for 30 min. In addition, the inhibitions by 1 mM Cu^2+^ and Cr^2+^ were 59.1% and 36.8%, respectively. It was noted that activity was enhanced by 21.4% and 31.2% after adding 1 mM and 5 mM Mg^2+^, respectively, and, thus, it was reasonable to assume that rPsRNR required a certain concentration of Mg^2+^ to improve its activity. Similarly, *E. coli* and *Mycoplasma genitalium* RNase R were found to be most active at 0.1–0.5 mM Mg^2+^ [7,27]. Besides, the activity of RNase H3 from *Aquifex aeolicus* was improved in the presence of Mg^2+^ [29]. Furthermore, 1 mM of ethylenediamine tetraacetic acid (EDTA) inhibited rPsRNR activity completely, which was similar to the report of RNase R from *E. coli* [7] and *Mycoplasma genitalium* [27].

### 2.6. Enzyme Kinetics of rPsRNR

Kinetics parameters *(K*_m_, *V*_m_, and *k*_cat_) of rPsRNR were measured at different temperatures (Figure 6). As shown in Table 3, the *K*_m_ values of rPsRNR decreased remarkably from 1.008 to 0.339 μM from 0 to 30 °C. The *K*_m_ trend of endonuclease I was the same as rPsRNR [30]. However, the *K*_m_ values of cold-adapted invertase increased with increasing temperatures, and *K*_m_ values were higher than its thermostable counterparts to decrease the activation free-energy barrier [31]. The *K*_m_ and *V*_max_ values of RNase HII from *Aeropyrum pernix* were 0.361 mM and 0.196 mM/min at 30 °C, respectively [28]. In addition, an extracellular ribonuclease from a *Bacillus* sp. exhibited a *K*_m_ value of 0.12 mg/mL and a *V*_max_ value of 55.5 μg/mL/min at 37 °C [32]. In addition, the *k*_cat_ values at 0 °C and 30 °C were 27.126 s^−1^ and 70.767 s^−1^, respectively. In generally, cold-adapted enzymes were quite efficient in compensating for the reduced reaction rates at low temperatures by improving of the *k*_cat_ value [33]. Therefore, it is speculated that rPsRNR, like endonuclease I from cold-adapted *Vibrio salmonicida* [30], might adapt to low temperatures by increasing *k*_cat_. Obviously, as the temperature increased, the *k*_cat_ gradually increased in a similar fashion to the tendency of cold-adapted β-d-galactosidase to *k*_cat_ at 10–30 °C [34]. Overall, the high catalytic efficiency at low temperatures of rPsRNR may make it a potential candidate for molecular biology applications.

### 2.7. Enzyme Thermodynamics of rPsRNR

Thermodynamic parameters (∆*H*, ∆*S*, and ∆*G*) of the rPsRNR were also measured (Table 4). The ∆*G* values of the rPsRNR increased from 59.20 to 63.55 kJ/mol when the temperature increased from 0 to 30 °C. This trend has been found in the xylanase rXynAGN16L from *Arthrobacter* sp. GN16 [35]. However, with the increase of temperature from 45 to 60 °C, the ∆*G* values of cold-adapted endoglucanase exhibited a downward trend, indicating that the thermostability of endoglucanase decreased with increasing temperature [36]. Furthermore, the ∆*H* values of the rPsRNR decreased from 0 to 30 °C. In general, the decrease of ∆*H* weakened the exponential temperature dependence of the reaction rate, thus contributing to the catalysis at low temperature. Therefore, rPsRNR showed smaller ∆*H* than psychrophilic endonuclease I and cold-adapted xylanase rXynAGN16L [30,35].

## 3. Materials and Methods 

### 3.1. Strains and Bacteria Cultivation

The RNase R-producing strain *Psychrobacter* sp. ANT206 (GenBank accession numbers MK968312) was isolated from Antarctic sea-ice (68° 300 E, 65° 000 S). Strain ANT206 was cultured in a 2216E medium at 12 °C and was shook at 200 rpm. pET-28(a) was stored in our lab and used as the vector for protein expression. *E. coli* BL21 was used as the host for protein expression. *E. coli* strains and recombinant bactera were cultured in a Luria Bertani (LB) medium containing kanamycin (100 mg/L).

### 3.2. Identification of psrnr Gene

According to the sequence and annotation of the genome of *Psychrobacter* sp. ANT206 (data not shown), the forward primers 5′-ACTGGATCCATGT CAAACCAAGATC-3′ (*Bam*HI site underlined) and the reverse primers 5′-TACCTCGAGCGCTCTTTTTACTACT-3′ (*Xho*I site underlined) were designed and amplified by PCR for the full-length RNase R gene (*psrnr*). The complete amino acid sequences of the PsRNR were obtained through an open reading frame finder (ORF finder). The catalytic sites were predicted by CD-Search (https://www.ncbi.nlm.nih.gov/Structure/cdd/wrpsb.cgi). Multiple sequence alignments were performed using the Bioedit and ESPript program (http://espript.ibcp.fr/ESPript/cgi-bin/ESPript.cgi). 

### 3.3. Homology Modeling of PsRNR

The 3D structure of the PsRNR was established by the SWISS-MODEL server and verified using SAVES V 5.0 (http://servicesn.mbi.ucla.edu/SAVES/) (University of California, Los Angeles, CA, USA). PyMOL 2.2.0 software (DeLano Scientific LLC, San Carlos, CA, USA) was used to visualize homology modeling with EcRNase R (PDB ID: 5XGU) as a template. In addition, salt bridges were predicted to use the VMD1.9.3 (University of lllinois, Urbana–Champaign, USA). Meanwhile, protein intramolecular interactions (hydrogen bonds, cation-pi interactions, hydrophobic interactions, and aromatic interactions) were predicted by the Protein Interactions Calculator program (http://pic.mbu.iisc.ernet.in/job.html).

### 3.4. Site-Directed Mutagenesis

Using the vector pET-28a (+) as a temple, the site-directed mutagenesis of T646, F654, L665, and H667 to Ala was performed with the QuikChange site-directed mutagenesis kit (Stratagene) by following the manufacturer’s protocol. The mutant primers were designed to amplify the DNA sequence, and the mutagenesis sites were underlined (Table 5).

### 3.5. Expression and Purification of PsRNR

In order to clone the *psrnr* gene and its mutant genes into the pET-28a(+) expression vector, the PCR product was inserted directly into pET-28a(+) vector. The recombinant plasmid was transformed into receptor *E. coli* BL21, and the sequences were then verified by sequencing. The recombinant strains grew in a LB medium and cultured at 37 °C until the OD_600_ nm was 0.6–0.8. After the optimization of protein expression, 238 mg/L IPTG was added for induction at 20 °C for 16 h. The induced cells were centrifuged and disrupted by ultrasonication (JY96-IIN, Shanghai, China). Then, the inclusion bodies were washed for removing the insoluble debris, treated with 8 M urea, and centrifuged at 7500× g for 15 min. The supernatant was diluted by 30 times by adding Tris-HCl (pH 8.0) at 25 °C for 2 h, which was the procedure of protein refolding. The protein solution was centrifuged at 12,000× g for 15 min, and the supernatant was the crude extract of proteins. Purification of rPsRNR and its mutant proteins were employed by Ni-NTA affinity chromatography (GE Healthcare, Uppsala, Sweden). The samples were eluted with 50 mM imidazole buffer (20 mM Tris-HCl, 500 mM NaCl, pH 8.0) at a flow rate of 0.5 mL/min. The purity and molecular mass of rPsRNR were determined by 12.0% polyacrylamide gels SDS-PAGE.

### 3.6. Enzyme Assay

The activity of rPsRNR toward yeast RNA was determined based on the previous method [37]. The reaction system contained a 0.1 M NaAc-HAc buffer (pH 6.0), 1.25 μM of yeast RNA, and 7.5 μg of purified protein. The reaction was performed for 10 min at specified temperatures and stopped by adding 350 μL ice–cold 3.4% perchloric acid. One unit of enzyme activity was defined as the amount of enzyme that brought about an increase in absorbance at 260 nm of 1 unit (U) per min in a cuvette of 1 cm path length of per mg of reaction system under the specified conditions. 

### 3.7. Biochemical Characteristics of rPsRNR

The purified protein (7.2 μg) was added in the reaction system to investigate its biochemical characteristics. To evaluate the optimal temperature of the rPsRNR, enzymatic activity was carried out at various temperatures (0–50 °C). For the enzymatic thermostability, purified rPsRNR was incubated at 40, 45, and 50 °C for 90 min. The optimum pH of rPsRNR was determined at 30 °C in NaAc/HAc (pH 4.0–6.0) and Na_2_HPO_4_/NaH_2_PO_4_ (pH 6.0–8.5) buffers. To test pH stability, purified rPsRNR was incubated in different pH buffers (4.0–8.5) at 30 °C for 30 min. To investigate the salt tolerance, purified rPsRNR was incubated at 30 °C for 30 min in 0–4.0 M NaCl buffers, and then the residual activity was assayed. The effects of various compounds on rPsRNR activity were investigated by the purified rPsRNR in buffer containing different reagents at 30 °C for 30 min, and the residual activity was assayed.

### 3.8. Kinetic Parameters and Thermodynamic Parameters of the rPsRNR

The reaction system of kinetic parameters contained 0.1 M of a NaAc–HAc buffer (pH 6.0), different concentrations of yeast RNA (0.25, 0.50, 0.75, 1.00, 1.25, and 1.5 μM), and purified rPsRNR (6.8 μg), all of which incubated at 0–30 °C for 10 min. Then, the reaction was stopped by adding 350 μL of ice-cold 3.4% perchloric acid. *K*_m_ and *V*_max_ were measured using the Michaelis–Menten equation [38]. The *k*_cat_ was calculated by the determination of kinetics, and the thermodynamic parameters (∆*H*, ∆*S,* and ∆*G*) were determined by the modification method [38].

## 4. Conclusions

A novel RNase R gene (*psrnr*) was cloned, expressed, and characterized in this study. PsRNR exhibited the structural characteristics of cold-adapted enzymes by homology modeling, including reduced hydrogen bonds and salt bridges. Furthermore, the residue of His667 was found to be necessary for the catalysis of PsRNR. After purification, rPsRNR exhibited maximal activity at 30 °C and pH 6.0, and it had excellent salt tolerance. Additionally, the thermodynamic characterization reflected the lower values of ∆*H*, ∆*S,* and ∆*G* at lower temperatures. Given these interesting characteristics, rPsRNR would be a potential candidate for antimicrobial treatment.

## Figures and Tables

**Figure 1 molecules-24-02229-f001:**
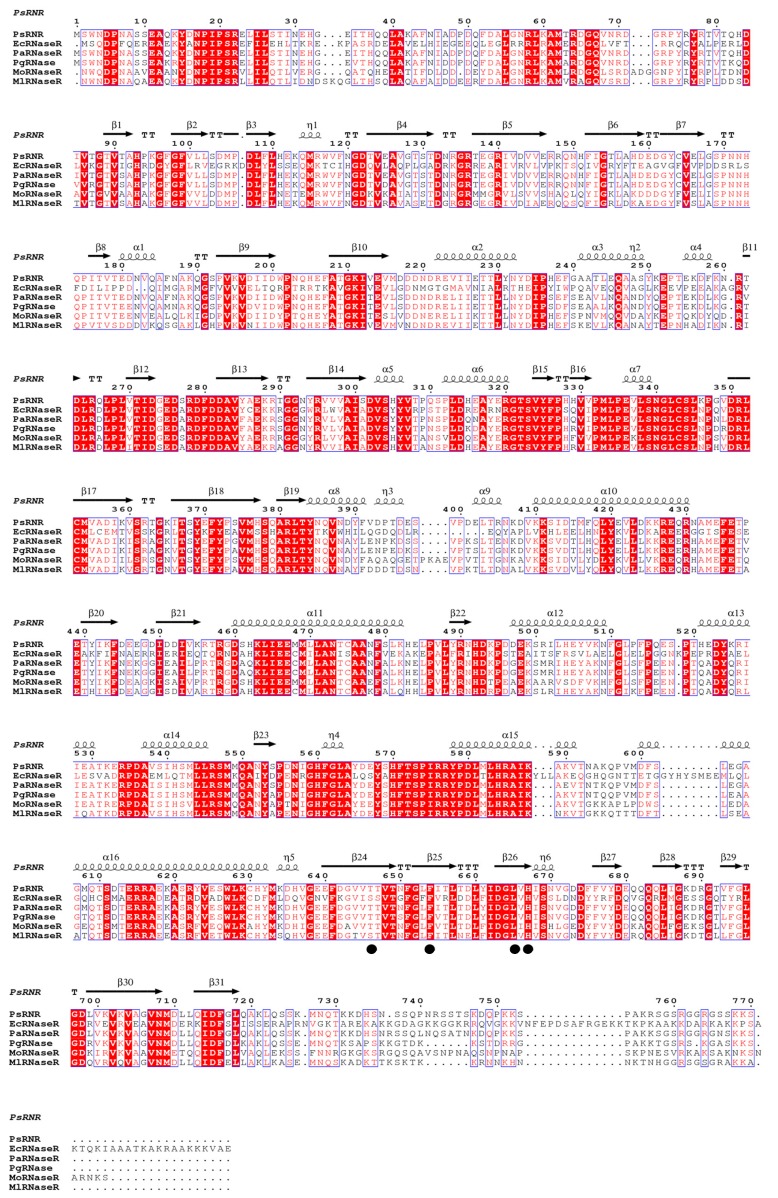
Alignment of the PsRNR sequence with other RNase R. The species names and GenBank accession numbers are as follows: PsRNR*, Psychrobacter* sp. ANT206 RNase R (MK624989), EcRNase R, *Escherichia coli* RNase R (WP_038432731, PDB ID: 5XGU), PaRNase R, *Psychrobacter arcticus* RNase R (WP_011281354), PgRNase R, *Psychrobacter glacincola* RNase R (WP_055125677), MoRNase R, *Moraxella osloensis* RNase R (WP_065264345), and MlRNase R, *Moraxella lincolnii* RNase R (WP_078306359). The conserved catalytic sites are indicated with cycles.

**Figure 2 molecules-24-02229-f002:**
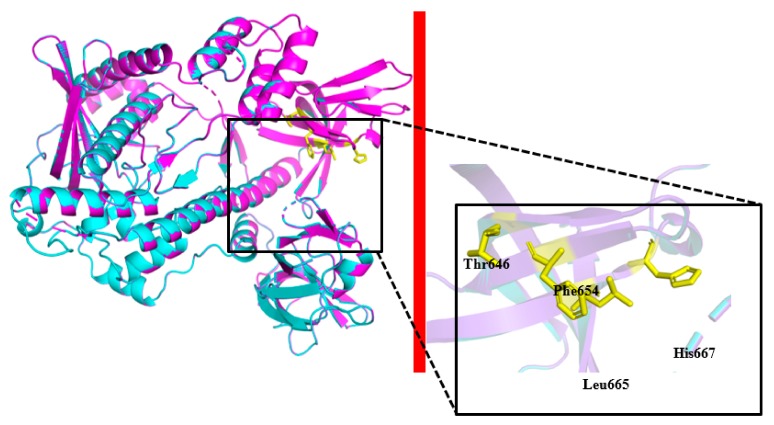
The 3D structure models of PsRNR and structure superimposition with EcRNase R (PDB ID: 5XGU). The catalytic triad residues are indicated as stick models.

**Figure 3 molecules-24-02229-f003:**
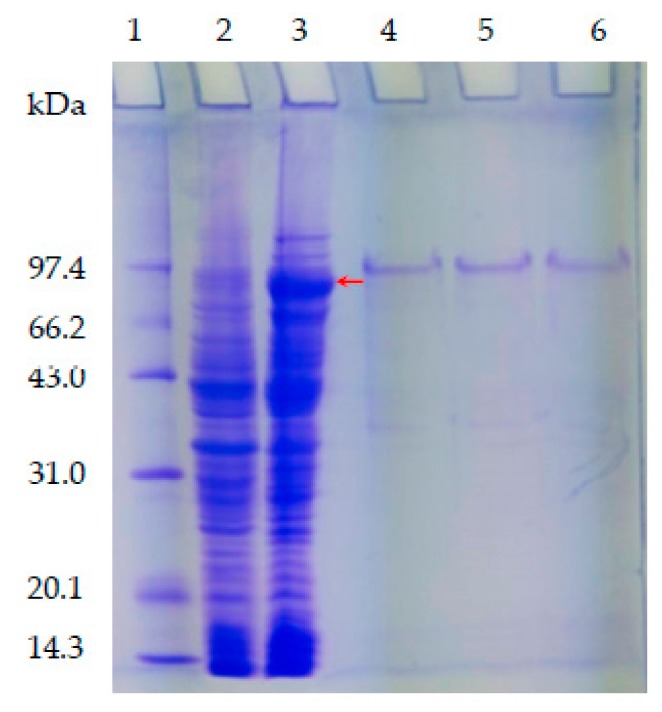
SDS-PAGE analysis of expression and purification of the rPsRNR. Lane 1, protein molecular weight marker; Lane 2, crude extract from the BL21/pET-28a(+)with IPTG induction; Lane 3, crude extract from the BL21/pET-28a(+)-rPsRNR; Lane 4–6, the purified rPsRNR protein by Ni-NTA using the buffer (pH 8.0) contained 20 mM Tris-HCl, 100 mM NaCl and 50 mM imidazole.

**Figure 4 molecules-24-02229-f004:**
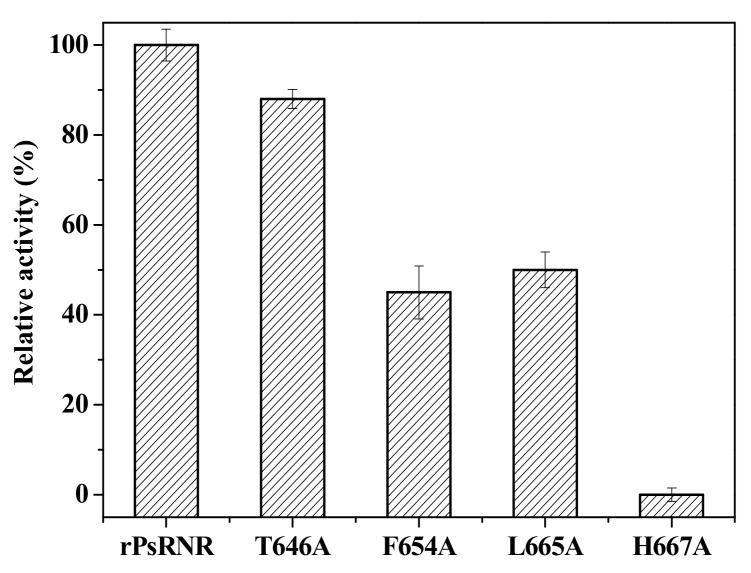
Measurement of the relative activity of the mutant rPsRNR generated by site-directed mutagenesis. The activity of the enzyme without mutation was defined as 100%.

**Figure 5 molecules-24-02229-f005:**
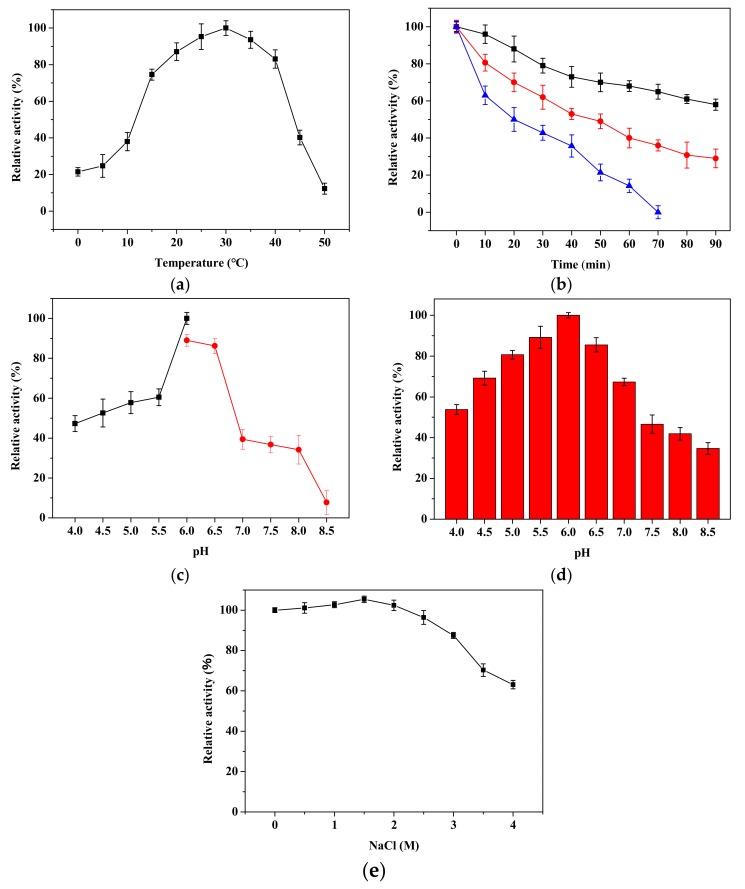
Biochemical characteristics of rPsRNR. (**a**) The optimal temperature was determined by measuring the activity at temperatures from 0 to 50 °C. Its maximal activity was taken as 100%. (**b**) Effect of temperatures on the stability of the purified rPsRNR. The enzyme was incubated at 40 °C (■), 45 °C (●), and 50 °C (▲), for 90 min. Its maximal activity was taken as 100%. (**c**) The optimal pH was determined by measuring the activity at pH from 4.0 to 8.5. The reaction contained NaAc/HAc(■) and Na_2_HPO_4_/ Na_2_HPO_4_ (●). (**d**) Effect of pH on the stability of the purified RNase R. The enzyme was incubated at 30 °C for 30 min. Its maximal activity was taken as 100%. (**e**) The enzyme was incubated by different concentrations of NaCl for 30 min. Its activity with 0 M NaCl was set as 100%.

**Figure 6 molecules-24-02229-f006:**
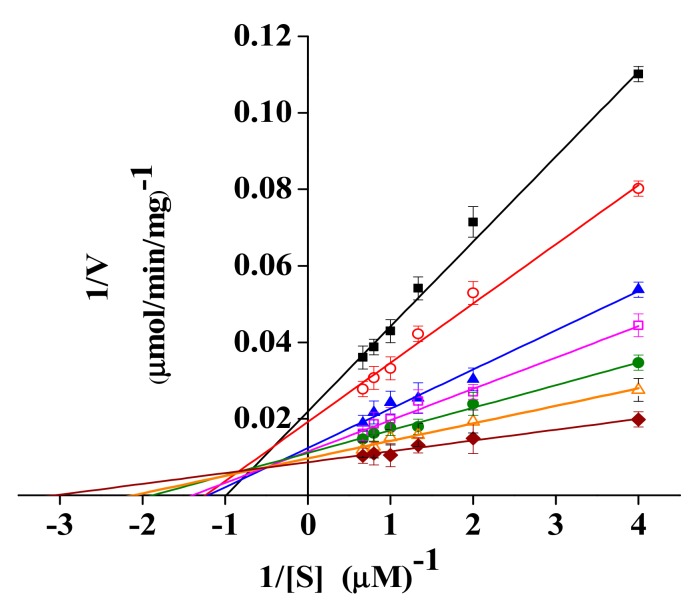
Lineweaver–Burk double reciprocal plots of purified RNase R with respect to yeast RNA at 0 °C (■), 5 °C (○), 10 °C (▲), 15 °C (□), 20 °C (●), 25 °C (△), and 30 °C (◆). Activity was measured as above described. Every experiment was done in triplicates.

**Table 1 molecules-24-02229-t001:** Comparison of structural adaption features between PsRNR and its homolog (EcRNase R).

Parameters	PsRNR	EcRNase R	Expected Effect on PsRNR
Salt Bridges	17	27	Stability
Hydrogen Bonds	478	492	
Cation-pi interactions	11	14	
Aromatic	22	23	
Hydrophobic interactions	447	543	Thermolability
Glycine residues	46	59	Flexibility
Proline residues	37	35	
Arginine residues	44	74	
Arg/(Arg + Lys)	0.47	0.56	

**Table 2 molecules-24-02229-t002:** Effect of various reagents on the rPsRNR activity.

Reagent	Concentration	Relative Activity (%)	Reagent	Concentration	Relative Activity (%)
None	--	100.0	None	--	100.0
Mg^2+^	1 mM	121.4 ± 2.6	Mg^2+^	5 mM	131.2 ±4.8
Ca^2+^	1 mM	102.3 ± 4.3	Ca^2+^	5 mM	86.0 ± 3.5
Zn^2+^	1 mM	81.8 ± 2.3	Zn^2+^	5 mM	75.8 ± 1.5
Fe^2+^	1 mM	90.9 ± 6.2	Fe^2+^	5 mM	85.9 ± 4.1
Cu^2+^	1 mM	40.9 ± 1.9	Cu^2+^	5 mM	20.5 ± 2.5
Pb^2+^	1 mM	72.7 ± 5.6	Pb^2+^	5 mM	65.3 ± 3.7
Cr^2+^	1 mM	63.2 ± 3.7	Cr^2+^	5 mM	62.7 ± 5.5
Ba^2+^	1 mM	2.1 ± 2.9	Ba^2+^	5 mM	ND
EDTA	1 mM	ND	EDTA	5 mM	ND

ND: Activity was not detected.

**Table 3 molecules-24-02229-t003:** The kinetics parameters of the rPsRNR.

Temperature (°C)	*V*_m_ (μmol/min/mg)	*K*_m_ (μM)	*k*_cat_ (s^−1^)
0	44.932	1.008	27.126
5	52.856	0.828	31.910
10	79.288	0.798	47.867
15	84.250	0.684	50.863
20	89.864	0.533	54.252
25	103.69	0.472	62.600
30	117.220	0.339	70.767

**Table 4 molecules-24-02229-t004:** The thermodynamic parameters of the rPsRNR.

Temperature (°C)	∆*H* (KJ/mol)	∆*S* (J/mol·K)	∆*G* (KJ/mol)
0	19.24	−146.30	59.20
5	19.20	−146.52	59.95
10	19.16	−144.66	60.12
15	19.12	−145.62	61.08
20	19.08	−146.50	62.02
25	19.03	−146.68	62.77
30	18.99	−146.99	63.55

**Table 5 molecules-24-02229-t005:** Primers used for mutations of PsRNR.

Mutagenesis	Primers Sequences
T646A	5′-CAAAGTTTGTGACGGTAGCTACGACACCATCGAACTC-3′
	5′-GAGTTCGATGGTGTCGTAGCTACCGTCACAAACTTTG-3′
F654A	5′-CAAATCCGTCAGAGTAATAGCTAAACCAAAGTTTGTGACGG-3′
	5′-CCGTCACAAACTTTGGTTTAGCTATTACTCTGACGGATTTG-3′
L665A	5′-GTTTGAGATATGCACCGCACCATCGATATACAAATCCG-3′
	5′-CGGATTTGTATATCGATGGTGCGGTGCATATCTCAAAC-3′
H667A	5′-CACCAACGTTTGAGATAGCCACCAAACCATCGATATAC-3′
	5′-GTATATCGATGGTTTGGTGGCTATCTCAAACGTTGGTG-3′

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
