# Peer review of "A Novel Cold-Adapted and Salt-Tolerant RNase R from Antarctic Sea-Ice Bacterium Psychrobacter sp. ANT206"

_molecules, 2019, doi:10.3390/molecules24122229_

Round 1

Reviewer 1 Report

This study presents the cloning, expression in E. coli and biochemical characterization of a novel RNase from Psychrobacter sp. ANT206. Finding an enzyme with intriguing and potentially useful biotechnological properties is a worthwhile goal. The new cold-adapted RNase is definitely an interesting subject. The manuscript provides basic data on enzyme isolation and characterization.

Some minor points are listed below:

Figure 1. Please correct figure legend (line 84).

Figure 2. Please revise it. Looks like some microorganism names have moved.

Line 95. “The rPsRNR with an approximately molecular mass of 91.4 kDa…” however the SDS-PAGE analysis shows a protein band of 97.4 kDa.

Line 171. It would be interesting to compare these kinetic parameters with those of others RNases.

Line 206. Provide the accession number.

Is all the protein of interest in the insoluble fraction? Have the authors optimized the expression of protein?

Therefore, I would like to recommend the editors to accept this manuscript with minor revision. If the authors are able to address the issues I have mentioned above the manuscript would be publishable.

Author Response

1. Figure 1. Please correct figure legend (line 84).

Response: We appreciate very much for the Reviewer’s good comments and kind recommendation. This part has been revised in this version, please see page 3 Figure 1.

2. Figure 2. Please revise it. Looks like some microorganism names have moved.

Response: We appreciate very much for the Reviewer’s comment. According to the third reviewer, Figure 2 was deleted in new vision.

3. Line 95. “The rPsRNR with an approximately molecular mass of 91.4 kDa…” 

however the SDS-PAGE analysis shows a protein band of 97.4 kDa.

Response: We appreciate very much for the Reviewer’s good comments and kind recommendation. After purification, the target protein is a single major band, and the SDS-PAGE shows upward drift, which is a normal phenomenon of SDS-PAGE. After multiple results, we calculated the molecular weight of recombinant protein based on the differential expression of inclusion bodies (Figure 3, lane 3, red arrow). This part has been revised. Please see page 2 line 79-82. Additionally, new Figure 3 in page 4 was replaced in new vision.

4. Line 171. It would be interesting to compare these kinetic parameters with those 

of others RNases.

Response: We appreciate very much for the Reviewer’s good comments. Because there are fewer reports on the kinetics of RNase R, this part has been added as much as possible. Please see page 7 line 177-180.

5. Line 206. Provide the accession number.

Response: We appreciate very much for the Reviewer’s comment. At present, the whole genetic data and analysis of Psychrobacter sp. ANT206 are underway, and other gene sequences of this strain are in a state of protection and cannot be published for the time being. In this version we provide the accession numbers of psrnr gene sequence and 16SrRNA sequence. Please see page 2 line 52 and page 8 line 208-209.

6. Is all the protein of interest in the insoluble fraction? Have the authors optimized 

the expression of protein?

Response: We appreciate very much for the Reviewer’s good comments and kind recommendation. In the preliminary experiment, we extracted the proteins in the supernatant solution and inclusion body and tested them by SDS-PAGE and activity assay. This part has been revised. Please page 9 line 241-242.

Therefore, I would like to recommend the editors to accept this manuscript with minor revision. If the authors are able to address the issues I have mentioned above the manuscript would be publishable.

Reviewer 2 Report

Minor revision:

1. Experimental tests' conditions has not been mentioned clearly in the manuscript.

2. Figure 4: The graphs trends are a little different. So, it is very ambiguous. It should be explain clearly. Please clarify it.

3. Introduction needs to improve the state of the art since denote lack of information and the references used are not always the best or the most adequate, or are obsolete

4. Results are very prolix. Please, avoid M&M and comments that must be moved to the corresponding sections.

Author Response

1. Experimental tests' conditions has not been mentioned clearly in the manuscript.

Response: We appreciate very much for the Reviewer’s useful suggestion. This part has been added. Please see page 8 line 208-209; page 9 line 220-221, 234-235, 241-242; page 10 line 254-256, 269-271.

2. Figure 4: The graphs trends are a little different. So, it is very ambiguous. It should be explain clearly. Please clarify it.

Response: We appreciate very much for the Reviewer’s good comments and kind recommendation. New Figure 3 in page 4 was replaced in new vision.

3. Introduction needs to improve the state of the art since denote lack of information and the references used are not always the best or the most adequate, or are obsolete

Response: We appreciate very much for the Reviewer’s useful suggestion. The whole introduction was modified in this version. Please see page 1 line 27-28, line 31-39.

4. Results are very prolix. Please, avoid M&M and comments that must be moved to the corresponding sections.

Response: We appreciate very much for the Reviewer’s useful suggestion. This part was modified in this version, please see page 5 line 114-115, 125, 129, 141-142, 146; page 7 line 173-174; page 8 line 195; page 9 line 234-235.

Reviewer 3 Report

This article describes the obtaining and characterization of a novel RNase R from Psychrobacter sp. including: (i) the identification of the gene, (ii) the creation of a 3D homology model and structure analysis based of the homology model, (iii) protein obtaining and purification; (iv) study of the mutation of the amino acids proposed be involved in the catalysis; (vi) general biochemical characterization (i.e. optimum temperature and pH, thermostability, effect of the pH on the stability and influence of different metal ions as well as their absence); and (vii) study of the kinetic parameters at different temperatures and enzyme thermodynamics. Unfortunately, I have to reject the acceptance of this manuscript. My decision is based on the following points:

-       The English used should be improved as there are way too many formal and linguistic mistakes, which make the text difficult to read and to understand. Thus, major rewriting/language editing would be needed.

-       The relevance and applicability of the results obtained is not clear to the reader. It should be stated how this research contributes to its field and why is it valuable.

-       Concerning the Results and Discussion section, generally it is hard to understand the results and discussions are poorly described. Some specific examples are:

·        The identification of the catalytic amino acids described in section 2.1. should clarify that those are proposed catalytic amino acids and why are they proposed. Additionally, the description of the catalytic amino acids described for other RNases does not give any further relevant information to the reader. Figure 2 is also unnecessary.

·        The long description of the creation of and structural model does not add any further value to the article since the target amino acids were already identified by sequence alignment.

·        Figure 4, the quality of the SDS-Page is very low, the wells of both cell free extracts were loaded with too much protein and the one with the purified protein with too little. Therefore it is difficult to see properly the protein bands.

·        The link for the comparison and discussion of the results previously obtain with other enzymes and those obtained in this work in not always clear.

·        The graphs obtained during the kinetic characterization of the enzyme should be included in the article, at least in the supplementary information.

-       Description of the experiments in Materials and Methods section is difficult to understand and loose, lacking important details in order to reproduce the experiments.

Author Response

1. This article describes the obtaining and characterization of a novel RNase R from Psychrobacter sp. including: (i) the identification of the gene, (ii) the creation of a 3D homology model and structure analysis based of the homology model, (iii) protein obtaining and purification; (iv) study of the mutation of the amino acids proposed be involved in the catalysis; (vi) general biochemical characterization (i.e. optimum temperature and pH, thermostability, effect of the pH on the stability and influence of different metal ions as well as their absence); and (vii) study of the kinetic parameters at different temperatures and enzyme thermodynamics. Unfortunately, I have to reject the acceptance of this manuscript. My decision is based on the following points:- The English used should be improved as there are way too many formal and linguistic mistakes, which make the text difficult to read and to understand. Thus, major rewriting/language editing would be needed.

Response: We appreciate very much for the Reviewer’s good comments and kind recommendation. RNase R could participate in an essential cell function and was crucial for RNA metabolism at low temperature. Due to the function of RNase R in RNA metabolism and biological activities, it has great application value in medicine and molecular biology research. However, there are few studies concerning the its structural and catalytic properties of RNase R from Antarctic organisms. Based on this, this paper carries on the related research. We revised the manuscript carefully to avoid the language errors. And we have consulted a professional English language editing services to check the English. And we believe that the language now is acceptable for the review process.

2. The relevance and applicability of the results obtained is not clear to the reader. It should be stated how this research contributes to its field and why is it valuable.

Response: We appreciate very much for the Reviewer’s good comments and kind recommendation. This part was revised. Please see page 5 line 140-141; page 7 line 186-187; page 10 line 279-281.

3. Concerning the Results and Discussion section, generally it is hard to understand the results and discussions are poorly described. Some specific examples are:

The identification of the catalytic amino acids described in section 2.1. should clarify that those are proposed catalytic amino acids and why are they proposed. Additionally, the description of the catalytic amino acids described for other RNases does not give any further relevant information to the reader. Figure 2 is also unnecessary.

Response: We appreciate very much for the Reviewer’s good comments and kind recommendation. This part has been added. Please see page 2 line 54-57. Additionally, Figure 2 has been deleted.

4. The long description of the creation of and structural model does not add any further value to the article since the target amino acids were already identified by sequence alignment.

Response: We appreciate very much for the Reviewer’s good comments and kind recommendation. Homology modeling is an efficient tool that has been widely used in the study of the cold-adaption mechanism of psychrophilic enzymes and their homologs [Hashim NHF, et al, Extremophiles, 2018, 22: 607-616; KS Siddiqui, R Cavicchioli, Annu. Rev. Biochem. 75 (2006) 403-433]. In this study, this tool was used to explain the molecular cold-adapted mechanism of PsRNR. Additionally, This part has been reduced.

5. Figure 4, the quality of the SDS-Page is very low, the wells of both cell free extracts were loaded with too much protein and the one with the purified protein with too little. Therefore it is difficult to see properly the protein bands.

Response: We appreciate very much for the Reviewer’s good comments and kind recommendation. The clearer figure of SDS-PAGE has been added in new vision. Please see page 4 Figure 3.

6. The link for the comparison and discussion of the results previously obtain with other enzymes and those obtained in this work in not always clear.

Response: We highly agree and appreciate very much for the Reviewer’s nice comments. Up to now, there have been few reports on the properties of RNase R, we have already cited the existing literature related to the properties of RNase R, and we have added as many literatures about other RNases as possible. Please see page 2 line 56-57; page 5 line 139-140; page 5-6 line 151-153; page 7 line 177-180.

7. The graphs obtained during the kinetic characterization of the enzyme should be included in the article, at least in the supplementary information.

Response: We appreciate very much for the Reviewer’s good comments and kind recommendation. This part has been added. Please see page 7 line 173-174 and Figure 6. 

8. Description of the experiments in Materials and Methods section is difficult to understand and loose, lacking important details in order to reproduce the experiments.

Response: We appreciate very much for the Reviewer’s useful suggestion. This part was modified in this version. Please see page 8 line 208-209; page 9 line 220-221, 234-235, 241-242; page 10 line 254-256, 269-271.

Round 2

Reviewer 3 Report

Although I appreciate the efforts made by the authors, in my opinion some there are still issues to be addressed before acceptance of the paper.

-        The text is still difficult to follow and to understand.

-        It would be interesting to state the relevance and applicability of these results in a more specific way than “potential candidate for medicine and molecular biology”.

-        Table 2: there should be a blank space between the figures and the units (mM).

-        In Materials and Methods:

Section 3.3: The protein used as template for the 3D structure built with SWISS-MODEL should be stated

Section 3.6: the quantity of enzyme added to those reactions should be described as Us, or at least mg of enzyme, not only as volume.

Section 3.7: Is not clear if the activity measurements described here are preformed exactly as previously described. If so description of the Us or mg of enzyme added would be also necessary, as the activity/protein recovery can drastically vary from one overexpression/purification batch to another.

Section 3.8: Reaction condition conditions for the kinetic experiments should be described (buffer, ph, enzyme concentration, presence of metal ions…).

Author Response

Responses to the Referees’ Comments

Comments to the Author:

Although I appreciate the efforts made by the authors, in my opinion some there are still issues to be addressed before acceptance of the paper.

1. The text is still difficult to follow and to understand.

Response: We appreciate very much for the Reviewer’s good comments and kind recommendation. We revised the manuscript carefully to avoid the language errors. And we have consulted a professional English language editing services to check the English. And we believe that the language now is acceptable for the review process.

2. It would be interesting to state the relevance and applicability of these results in a more specific way than “potential candidate for medicine and molecular biology”.

Response: We appreciate very much for the Reviewer’s good comments and kind recommendation. This part was revised. Please see page 1 line 17-18, 38-40; page 10 line 286.

3.Table 2: there should be a blank space between the figures and the units (mM).

Response: We appreciate very much for the Reviewer’s good comments and kind recommendation. This part was revised. Please see page 7 Table 2.

4. In Materials and Methods:

Section 3.3: The protein used as template for the 3D structure built with SWISS-MODEL should be stated

Response: We appreciate very much for the Reviewer’s good comments and kind recommendation. This part was added. Please see page 9 line 228.

5. Section 3.6: the quantity of enzyme added to those reactions should be described as Us, or at least mg of enzyme, not only as volume.

Response: We appreciate very much for the Reviewer’s good comments and kind recommendation. This part was added. Please see page 10 line 256-257.

6. Section 3.7: Is not clear if the activity measurements described here are preformed exactly as previously described. If so description of the Us or mg of enzyme added would be also necessary, as the activity/protein recovery can drastically vary from one overexpression/purification batch to another.

Response: We appreciate very much for the Reviewer’s good comments and kind recommendation. The activity measurements described here are preformed exactly as previously described. And this part was added. Please see page 10 line 262-263.

7. Section 3.8: Reaction condition conditions for the kinetic experiments should be described (buffer, ph, enzyme concentration, presence of metal ions…).

Response: We appreciate very much for the Reviewer’s good comments and kind recommendation. This part was added. Please see page 10 line 273-276.